# In Silico Optimization of Mass Spectrometry Fragmentation Strategies in Metabolomics

**DOI:** 10.3390/metabo9100219

**Published:** 2019-10-09

**Authors:** Joe Wandy, Vinny Davies, Justin J. J. van der Hooft, Stefan Weidt, Rónán Daly, Simon Rogers

**Affiliations:** 1Glasgow Polyomics, University of Glasgow, Glasgow G61 1BD, UK; joe.wandy@glasgow.ac.uk (J.W.); stefan.weidt@glasgow.ac.uk (S.W.); ronan.daly@glasgow.ac.uk (R.D.); 2School of Computing Science, University of Glasgow, Glasgow G12 8RZ, UK; vinny.davies@glasgow.ac.uk; 3Bioinformatics Group, Department of Plant Sciences, Wageningen University, Wageningen 6780 PB, The Netherlands; justin.vanderhooft@wur.nl

**Keywords:** liquid chromatography–mass spectrometry (LC/MS), fragmentation (MS/MS), data-dependent acquisition (DDA), simulator, in silico

## Abstract

Liquid chromatography (LC) coupled to tandem mass spectrometry (MS/MS) is widely used in identifying small molecules in untargeted metabolomics. Various strategies exist to acquire MS/MS fragmentation spectra; however, the development of new acquisition strategies is hampered by the lack of simulators that let researchers prototype, compare, and optimize strategies before validations on real machines. We introduce Virtual Metabolomics Mass Spectrometer (ViMMS), a metabolomics LC-MS/MS simulator framework that allows for scan-level control of the MS2 acquisition process in silico. ViMMS can generate new LC-MS/MS data based on empirical data or virtually re-run a previous LC-MS/MS analysis using pre-existing data to allow the testing of different fragmentation strategies. To demonstrate its utility, we show how ViMMS can be used to optimize *N* for Top-N data-dependent acquisition (DDA) acquisition, giving results comparable to modifying *N* on the mass spectrometer. We expect that ViMMS will save method development time by allowing for offline evaluation of novel fragmentation strategies and optimization of the fragmentation strategy for a particular experiment.

## 1. Introduction

Liquid chromatography (LC) tandem mass spectrometry (MS/MS) is commonly used to identify small molecules in untargeted metabolomics. In this setup, chemicals elute through the liquid chromatographic column at different retention times (RTs) before entering the mass spectrometer and potentially undergoing fragmentation. Fragmentation produces distinct patterns of fragment peaks at different mass-to-charge ratios (*m*/*z*s) that can be used to annotate chemical structures [1,2]. The choice of fragmentation strategy, which determines how precursor ions are selected for further fragmentation in tandem mass spectrometry, is an important factor affecting the coverage and quality of MS/MS spectra available for subsequent analysis. Many strategies exist to perform fragmentation, including data-independent acquisition (DIA) and data-dependent acquisition (DDA), and new and improved fragmentation strategies are constantly being introduced [3,4]. However, evaluating and comparing different strategies is challenging since the chemicals present in the samples in untargeted metabolomics studies are generally unknown, making it hard to judge whether a certain strategy leads to optimal MS/MS coverage. Currently, this is usually done by trying different fragmentation settings on the instrument followed by manual inspection for the samples of interest.

An appealing alternative way to evaluate fragmentation strategies is using a simulator, which can replicate the underlying LC-MS/MS processes and allow researchers to prototype and compare strategies before validation on the actual MS instrument. Although some mass spectrometry simulators exist they are typically focused on proteomics and do not include simulation of the MS2 acquisition strategy within a chromatographic run [5,6,7,8,9,10]. Additionally, existing simulators do not allow for real-time control of scan events (such as programmatically determining which *m*/*z* ranges to scan at a particular retention time), a crucial function for developing novel fragmentation strategies that can be controlled through libraries available with modern mass spectrometers, e.g., using the Instrument Application Programming Interface (API) available for Thermo Tribrid instruments [11] that has begun to generate interest within the mass spectrometry community (e.g., [12]).

In this work, we introduce Virtual Metabolomics Mass Spectrometer (ViMMS) a modular LC-MS/MS simulator for metabolomics that allows for real-time scan-level control of the MS2 acquisition process in silico. ViMMS works by creating a set of chemical objects, each with its own chromatogram, RT and intensity, fragmentation spectra and propensity to generate particular adducts. These can be created from a list of known metabolites (for example from the Human Metabolome Database, HMDB [13]) or from chromatographic peaks extracted in experimental .mzML files. A selection of controllers that implement different fragmentation strategies are available, including standard Top-*N* strategies but also MS1-only simulation as they also form a part of LC-MS/MS experiments. Using the appropriate controllers, users can benchmark and test different strategies and obtain simulated results in mzML format (the entire simulator state can also be saved for inspection later).

The idea of ViMMS is to offer the functionality of simulating MS1 and MS2 generation processes, but also to be modular enough that additional features are easily integrated in the framework. The development of a simulator such as ViMMS makes it possible to optimize MS/MS acquisition in silico without having to use valuable instrument time. Please note that our proposed tool is not a new acquisition strategy itself, but rather a framework that makes the development, testing, and benchmarking of such acquisition strategies easier. All code and examples are available at https://github.com/sdrogers/vimms and demonstrate how our simulator framework can be used in an interactive setting via Jupyter Notebooks. We demonstrate the utility of ViMMS with two examples: first we perform an experiment to vary *N* (the number of precursor peaks selected for fragmentation in standard Top-*N* DDA fragmentation strategy) in silico as well as the dynamic exclusion window (DEW) that is used to exclude ions from fragmentation for a certain time and evaluate how changing those parameter settings affects fragmentation coverage and the quality of MS1 peak picking. We then validate these results by comparing the ViMMS output against experimental data when both *N* and dynamic exclusion window parameters are varied. Secondly, we use the simulator to reproduce key results from a novel fragmentation strategy, data-set-dependent acquisition (DsDA) [4], demonstrating how ViMMS can be used to compare fragmentation strategies before implementation in an actual MS instrument.

## 2. Materials and Methods

### 2.1. LC-MS/MS Materials and Methods

#### 2.1.1. Samples

Beer and urine samples (labelled *multi-beer* and *multi-urine*) from a previously published study [14] are used in our experiments. Here we briefly summarize the sample preparation and analytical platform for the *multi-beer* and *multi-urine* in [14]. 19 different beers were collected from bottles over a period of 5 months and frozen immediately after sampling. 22 urine samples were obtained from a clinical trial of an anonymized cohort of elderly hypertensive patients who were administered several drugs, including antihypertensives. 5 μL of beer/urine was extracted in 200 μL of chloroform/methanol/water (1:3:1) at 4 ${}^{\circ }$C, vortexed for 5 min at 4 ${}^{\circ }$C and centrifuged for 3 min (13,000 g) at 4 ${}^{\circ }$C. The resulting supernatant was stored at −80 ${}^{\circ }$C until analysis, and a pooled aliquot of the 22 selected urine samples and 19 beer samples were prepared prior to LC-MS/MS runs.

On top of the existing *multi-beer* and *multi-urine* samples, we also introduce newly generated beer data in this study. One beer extract (labelled *BeerQCB*) was selected for repeated and reproducible sampling across this experiment. An English premium bitter (Black Sheep Ale, 4.4 %) was purchased from a local supermarket. Beer metabolites were extracted by addition of chloroform and methanol to the ratio of 1:1:3 (*v*/*v*/*v*), as previously described, except for the total volume being scaled up to 100 mL. The solution was thoroughly mixed using a vortex mixer, before protein and other precipitates removed by centrifuging at 14,000 rpm at 4 ${}^{\circ }$C for 10 min. The supernatant was removed, and aliquots stored at −80 ${}^{\circ }$C until needed.

#### 2.1.2. Liquid Chromatography

All samples underwent liquid chromatography separation under the following experimental conditions: a Thermo Scientific UltiMate 3000 RSLC liquid chromatography system was used for HILIC separation with a SeQuant ZIC-pHILIC column using a gradient elution with (A) 20 mM ammonium carbonate and (B) acetontrile. 10 μL of each sample was injected onto the column with initial conditions of 80% (B). A linear gradient from 80% to 20% (B) over 15 min, a wash of 5% (B) for 2 min, before re-equilibration at 80% (B) for 7 min (QE) or 9 min (Fusion). A constant flow rate of 300 μL/min was used. The column oven was maintained at a constant temperature of 25 ${}^{\circ }$C (QE) or 40 ${}^{\circ }$C (Fusion). Blank runs, quality control samples and three standard mixes were prepared according to the standard procedures at Glasgow Polyomics [14,15].

#### 2.1.3. Mass Spectrometer Acquisition

The *multi-beer* and *multi-urine* datasets were acquired using a Q-Exactive orbitrap mass spectrometer for LC-MS/MS. All full-scan spectra were acquired in positive ion mode only, with a fixed resolution of 70,000, with mass range 70–1050 *m*/*z*. Ions were isolated with 1.0 *m*/*z* width and fragmented with stepped HCD collision energy of 25.2, 60, 94.8% for both positive and negative ion modes. Fragmentation spectrum were acquired with the orbitrap mass analyzer with resolution of 17,500. Top 10 ions with an intensity threshold ≥1.3E5 were selected for fragmentation and then added to a dynamic exclusion window for 15 s. For more mass spectrometer acquisition details, we refer to [14].

The new validation dataset (*BeerQCB*) to simulate fragmentation performance in Section 3.4 was generated using an orbitrap fusion tribrid-series mass spectrometer. All full-scan spectra were acquired in positive ion mode only, with a fixed resolution of 120,000, with mass range 70–1000 *m*/*z*. To investigate the instrument performance with differing Top-*N* and dynamic exclusion windows, filters such as intensity threshold and monoisotopic peak determination were not used. This allowed for a consistent number tandem MS scans to be acquired under varying Top-*N* parameters and dynamic exclusion window (DEW), for N=(1,2,3,4,5,10,15,20,35,50) and DEW=(15,30,60,120). Ions were isolated with 0.7 *m*/*z* width and fragmented with fixed HCD collision energy of 25%. Fragmentation spectrum were acquired with the orbitrap mass analyzer with resolution of 7500.

#### 2.1.4. Data Transformation

Raw files from acquisition were converted into mzML format using MSconvert (Proteowizard). In the evaluation of the Top-*N* controller in Section 3.2 and Section 3.3, two of the *multi-beer* samples (labelled *multi-beer-1* and *multi-beer-2*) and two of the *multi-urine* samples (labelled *multi-urine-1* and *multi-urine-2*) are used. In Section 3.5 to evaluate DsDA, all *multi-beer* and *multi-urine* samples are used. Section 3.4 uses only the *BeerQCB* samples.

### 2.2. Computational Methods

#### 2.2.1. Overall Framework

The overall schematic for ViMMS can be found in Figure 1. ViMMS works by first creating *chemical* objects which represent the possible metabolites in a sample (Section 2.2.2). These objects contain information which defines how each chemical appears when scanned by the virtual mass spectrometer (yellow box in Figure 1). To get the information to fill the chemical objects, we create a database of spectral features from experimental data from which we can sample (Section 2.2.3). Regions of interest (ROIs) representing groups of mass traces that could potentially form chromatographic peaks are also extracted from experimental files and assigned to chemicals (Section 2.2.4). Unlike other simulators, e.g., [5,6,7,8,9], chemical objects can also be associated with fragment spectra that could themselves be extracted from spectral databases or generated using *in silico* fragment prediction methods (Section 2.2.5). *In silico* scan simulation in ViMMS (yellow box in Figure 1) proceeds as follows. A virtual mass spectrometer takes the list of chemical objects as input and generates MS1 and MS2 scans at appropriate RTs (Section 2.2.6). Scan parameters are determined by a controller that implements a particular fragmentation strategy. (Section 2.3). The proposed framework is designed to be completely modular such that a variety of situations and different fragmentation strategies can be tested. Finally, using the psims library [16] simulated results can be written as mzML files for further analysis in other tools. The entire state of the simulator over time can also be saved for inspections using the built-in pickle function in Python.

#### 2.2.2. Chemical Objects

Chemical objects in ViMMS can be created in two ways—by sampling chemical formulae from a relevant database and then associating them with chromatographic peaks, or by re-running an existing analysis. In the first method, formulae are first sampled from a metabolite database such as HMDB [13] (Synthetic Sample Workflow in Figure 1). Each chemical is given a starting RT (the first RT at which they will appear when scanned) and a maximum intensity value by sampling from the spectral feature database in Section 2.2.3. Based on the spectral information they contain, the chemical objects are able to generate MS1 peaks for the relevant adducts and isotopes, with the intensity of the chemical object split between the various adduct and isotope combinations and the *m*/*z* values being calculated based on the chemical’s assigned formula. Distributions over adduct intensities can be specified by the user. Finally, an ROI with a similar maximum intensity to the chemical is chosen (Section 2.2.4), and fragment peaks assigned (Section 2.2.5). It is also possible to generate multiple related samples of chemicals, whether biological or technical replicates. To do this we introduce independent Gaussian noise to the maximum intensity values and allow chemicals to be excluded from samples with a certain dropout probability.

When real data is available and the user wishes to re-run the same data under different fragmentation strategies, chemicals can also be extracted from an existing mzML file (Existing Sample Workflow in Figure 1). Here ROIs in the file are extracted and converted to chemical objects (we make the simplifying assumption that each ROI corresponds to a single unknown chemical). Unknown chemicals created in this manner will generate a single trace in the output (as opposed to multiple traces where multiple adducts and isotopes are generated.

#### 2.2.3. Spectral Feature Database

To generate data, we create a database of spectral features extracted from actual experimental data. This database is used to sample the features associated with a chemical including the *m*/*z*, RT, and maximum intensity values of observed MS1 and MS2 peaks, as well the number of fragment peaks found for typical scans. During simulation, the database is also used by the controller to sample for the duration of each scan (Section 2.2.6). To construct this database, users provide their data in mzML format. pymzML [17] is then used to load the input mzML files, extract the necessary features and construct the database which is stored as Python pickled format. In the case of Synthetic Sample Workflow in Figure 1, the database also stores information on the small molecules extracted from an external metabolite database such as HMDB.

#### 2.2.4. ROI Extraction and Normalization

ROIs are extracted using our Python [18] re-implementation of the ROI extraction procedure of XCMS’s CentWave algorithm [19] originally available in the R programming language [20], although ROIs could have easily been extracted with alternative software such as MZmine [21]. First, spectra in an mzML file are loaded using pymzML. Then the ROI extraction algorithm loops over all scans, extracting the raw traces (recorded in centroid mode) from observed spectra. This results in a list of peak features of (*m*/*z*, RT, intensity) values. Features are first filtered to remove any that have an intensity below some user-defined threshold. The current *m*/*z* value is matched to find existing ROIs that it could fall into within a mass tolerance window, defined as the window above and below the mean *m*/*z* of the ROI. If no match exists, then this feature forms its own ROI and gets added to the list of existing ROIs. ROIs that are not added to are closed and put aside. ROIs that contain fewer data points than a user-defined threshold parameter are discarded. Finally, ROIs are normalized so their *m*/*z* values are centered around 0, RT values start at 0, and intensity values are scaled to have a maximum of 1, such that they can be assigned to chemicals.

#### 2.2.5. MS2 Scan Generation

The MS2 scan generation process in ViMMS is modular and allows for different methods to be selected for generating and associating MS2 fragments to chemical objects. In our current implementation, two baseline methods are provided. The first is to assign *m*/*z* and intensity values to fragment peaks by randomly sampling from the spectral feature database in Section 2.2.3. This works for experiments where we can make the simplifying assumptions that fragment peaks are completely independent across scans. To reflect a more realistic scenario where groups of fragment peaks may co-occur in multiple fragmentation spectra [22], we provide a second method of assigning MS2 peaks in a fragmentation scan by following a truncated Chinese Restaurant Process (CRP) [23]. This allows for a fragment peak to have a greater likelihood to be selected again if it has been selected before in previous scans. The truncated CRP process follows the standard process of a CRP, but prevents the same MS2 peak being assigned to the same fragmentation spectra more than once. The modular nature of ViMMS means that it would be straightforward to incorporate MS2 prediction methods such as CFM-ID [24,25] or NEIMS [26].

#### 2.2.6. Scan Time

For accurate simulation of duty cycles, we sample scan durations of MS1 and MS2 scans from the spectral feature database in Section 2.2.3. Based on the MS level of the previous scan, as well as that of the scan about to be undertaken, the time for the scan about to take place is drawn from the times of those scans in the database which represent the relevant scan transition. The only time that this is not the case is when the DsDA controller is used (Section 2.3.3) as we have a fixed timing schedule. Scan times sampled in this manner will almost always not correspond to values observed in the original files from which the ROIs were extracted. This causes some difficulty with determining the intensity and *m*/*z* values of the chemicals that would be observed at this time, as they will not have previously been observed. To overcome this, we use a simple interpolation scheme (the trapezium rule) between the two nearest scans, which gives us estimates of the intensity and *m*/*z* values that would be expected for any chemical object at the previously unobserved RT.

### 2.3. Controllers

ViMMS is designed to be flexible, and to achieve this aim, we separate the simulation of mass spectrometer (generating spectra from chemicals) and the fragmentation strategy (determining which precursors to fragment) in the framework. Generating spectra from chemicals is implemented inside a virtual mass spectrometer, while different fragmentation strategies are implemented as controllers. To simulate a scan, the virtual MS iterates through chemical objects that each generate MS1 or MS2 peaks depending on the current RT and the MS level requested by the controlled. The virtual MS is also responsible for broadcasting events, such as when a new scan is generated or when acquisition is started or has been finished. Controllers can subscribe to these events and act upon them, for example by directing the virtual MS to perform different scans according to the current fragmentation strategy (yellow box in Figure 1). It is relatively straightforward to implement various controllers that perform different fragmentation strategies. Each controller is designed such that it is separate from the virtual mass spectrometer, allowing controllers to interact with either the virtual MS or with an actual MS instrument through an application programming interface as a future work.

#### 2.3.1. MS1 Controller

The MS1 controller is designed to replicate the process of generating MS1 full scans by a mass spectrometer. Given a start and end RT range, the MS1 controller steps through time and generates scans from chemicals. A scan therefore consists of *m*/*z* and intensity pairs for those chemicals that are currently eluting. The timings of the scans are determined based on experimental data by sampling from the spectral feature database, as described in Section 2.2.6. Scan results can be exported as an mzML file and viewed in standard programs such as TOPPView [27].

#### 2.3.2. Top-*N* DDA Controller

The Top-*N* controller performs standard DDA acquisition. In each duty cycle, the controller first performs an MS1 scan to establish the most intense precursor ions, followed by up to *N* fragmentation scans depending on the number of precursor ions selected for further fragmentation. To generate fragmentation scans, the Top-*N* precursor ions (in descending order of intensities) in the initial MS1 scan are isolated and fragmented. A dynamic exclusion window (DEW) is used to prevent precursor ions that have recently been analyzed from being fragmented again. In the controller, we also provide a threshold on the minimum MS1 intensity for a precursor ion to be selected for fragmentation.

#### 2.3.3. DsDA Controller

The DsDA [4] controller attempts to optimize fragmentation strategy over several similar samples. DsDA keeps track of which precursor ions have been fragmented in previous samples, and prioritizes those that have high MS1 intensity and have either not been fragmented, or have been fragmented producing low quality MS/MS spectra.

Implementing the full DsDA analysis pipeline in ViMMS requires the following process. First the DsDA controller, written in Python, calls the Top-*N* controller to perform an initial DDA analysis (for the first sample) using a fixed timing schedule. Once the initial DDA analysis is complete, the resulting mzML file is analyzed using the original DsDA scan prioritization algorithm written in R (available from https://github.com/cbroeckl/DsDA). This involves picking peaks and comparing the picked peaks to what has previously been fragmented. This information is used to determine at what *m*/*z* and RT locations new fragmentation scans should be performed. The prioritization algorithm attempts to get the highest quality MS/MS spectra for as many different precursor ions as possible. To avoid missing novel precursor ions that may not have appeared before, DsDA also includes an option called ‘MaxDepth’ which increases the probability of sampling rare features that the prioritization algorithm was originally designed to devalue. The resulting schedule is used for the analysis of the next sample using the Python-based DsDA controller, a process that is automatically repeated until all the samples have been analyzed.

## 3. Results

### 3.1. MS1 Simulations

To demonstrate the ability of ViMMS to simulate MS1 scans generated by chemicals from a metabolite database, we create a sample consisting of 6500 chemicals from HMDB and use the 19 full-scan experimental beer data from the *multi-beer* dataset to generate the spectral feature database (Synthetic Sample Workflow in Figure 1). The MS1 controller (Section 2.3.1) in ViMMS is used to perform a full-scan MS1 simulation. Simulation results are exported as an .mzML file and loaded into Jupyter Notebook for further analysis (all example notebooks can be found in our code repository).

Figure 2 shows examples of snapshots of full-scan chromatograms in TOPPView [27] for the actual experimental *multi-beer-1* sample (Figure 2A) and a simulated sample created in ViMMS (Figure 2B). The resulting spectra show similar characteristics to each other in terms of the shapes of the peaks and how they are observed in a full-scan samples. Individually the peaks appear at the different locations and with different profiles as a result of the simulation process, with the aim here not to directly copy the real beer sample, but create a sample with similar overall properties. A further demonstration of the similarity of the samples can be seen in boxplots of the XCMS picked peaks characteristics (RT, *m*/*z*, log intensity) shown in Appendix A. A user could also produce similar results with alternative peak picking algorithms such as MZmine.

### 3.2. Top-N Simulations

We now show an example of using the Top-*N* controller, available from ViMMS (described in Section 2.3.2). This controller accepts as input a list of chemicals objects and performs MS2 fragmentation simulation by isolating precursor (MS1) ions and producing scans containing product (MS2) peaks. To check that our Top-*N* simulation processes reflect reality, we conduct an experiment where existing chromatographic peaks from the *multi-beer-1* fragmentation file are loaded into the simulator (Existing Sample Workflow in Figure 1). Top-10 DDA fragmentation is performed using the Top-*N* controller and the resulting output compared to the original input file. The aim here is to assess how much our simulated file differs from the actual fragmentation file given the same input ROIs and similar fragmentation parameters (*N*=10, DEW=15 s). A visual snapshot of resulting spectra in TOPPView can be found in Appendix A and a comparison of when and where the fragmentation events occurred can be seen in Appendix A.

Figure 3a shows the number of MS1 and MS2 scans completed over time for the true and simulated scenarios. The total number of scans is very similar in both cases, as can also be seen in Appendix A. Figure 3b shows that the situations in which the simulator and actual data do not match typically involve low intensity precursors. Investigating the differences between the simulation and the real data in detail, we observe what seems to be unpredictable behavior from the mass spectrometer. For example, in some cases it fragments 9 instead of 10 ions (even when other ions are present above the minimum intensity that should not be excluded due to a previous fragmentation event), and on some occasions it fragments ions despite them being below the minimum intensity threshold. These differences might be due to our handling of the data in centroid mode (and the real MS controller operating in profile mode), and there will also be a small difference due to our randomly sampled scan times. Overall, however, we are confident that the behavior of the simulator is close enough to reality and that our Top-*N* controller captures the most important fragmentation events and can be used for further experiments in subsequent sections.

### 3.3. Varying *N* in Top-N Simulations

Choosing *N* in DDA is a critical part of method development. Increasing *N* ought to give better MS/MS coverage as more ions are fragmented. However, increasing *N* too far will result in many ions being fragmented below their minimum intensity threshold (even if they were above the minimum during the initial MS1 scan). In addition, larger *N* reduces the frequency of MS1 scans, which will have a detrimental effect on MS1 peak picking. ViMMS allows us to objectively investigate this trade-off, providing a strong evidence base for method development.

Consider a typical scenario where within an experimental batch, only Top-*N* DDA is performed and no full-scan data are available (an alternative scenario where both full-scan and Top-*N* data are acquired is also considered in Appendix A). In this case, it is standard to use only peaks picked from the MS1 scans (which we call *MS1 features*) in the DDA fragmentation files for further analysis. As already mentioned, increasing *N* could result in greater fragmentation coverage since more precursor ions are fragmented but also potentially fewer MS1 features from the fragmentation file due to fewer MS1 data points available for peak picking. Evaluating the best Top-*N* parameter that results in an optimal trade-off between fragmentation coverage and peak picking performance can be challenging on real data, but it is possible in a simulated environment such as ViMMS.

To perform this simulated experiment, first an existing full-scan file is loaded into ViMMS. The Top-*N* DDA controller (Section 2.3.2) can be run with a variety of different *N*s and the results evaluated. Based on these results we can choose the best *N* for future experiments on similar samples for that mass spectrometer. Given actual experimental full-scan MS1 files, the effect of varying *N* to simulated fragmentation coverage and peak picking quality can be evaluated with respect to the ground truth MS1 features found in both the full-scan and fragmentation files. For evaluation, the following definition of positive and negative instances (illustrated in Figure 4) is proposed:

**True Positives (TP)**: MS1 features from ground truth (found in both fragmentation and full-scan files) that are fragmented above the minimum intensity threshold.**False Positives (FP)**: MS1 features not from ground truth (found in fragmentation file but not in full-scan file) that are fragmented above the minimum intensity threshold.**False Negatives (FN)**: MS1 features not from ground truth (not found in fragmentation file but found in full-scan file) that are not fragmented or fragmented below the minimum intensity threshold.**True Negatives (TN)**: MS1 features not from ground truth (found in fragmentation file but not found in full-scan file) that are not fragmented or fragmented below the minimum intensity threshold.

It is worth noting that this evaluation strategy uses picked peaks as a ground truth. Peak picking is a process known to not be entirely accurate, although we believe that this represents a meaningful evaluation metric given the widespread use of peaking picking in metabolic analyses.

In our experiment, four existing Top-10 DDA files from the *multi-beer* and *multi-urine* samples are loaded into ViMMS using the Existing Sample Workflow in Figure 1. For each sample (labelled *multi-beer-1*, *multi-beer-2*, *multi-urine-1* and *multi-urine-2* respectively), DDA fragmentation is simulated using the Top-*N* controller in ViMMS. The parameter *N* for Top-*N* is varied in the range N=(1,2,3,4,5,6,7,8,9,10,15,20,25,…,100) in the simulator, while other parameters are fixed following Section 3.2. In this experiment we also fix the dynamic exclusion window (DEW) to 15 s and the minimum MS1 intensity to fragment to $1.75\times 10^{5}$ based on the actual parameters that were used to generate the data. Our results are evaluated in terms of precision, recall, numbers of peaks picked and $F_{1}$ score (Precision=TP/(TP+FP), Recall=TP/(TP+FN), $F_{1}=\left(2*Precision*Recall\right)/\left(Precision+Recall\right)$). To obtain the ground truth for evaluation, we performed peak picking using XCMS’ CentWave on both the full-scan and simulated fragmentation files using the parameters in [14].

Using the simulator, we observe that increasing *N* produces an initial increase followed by a decrease in precision (Figure 5a), suggesting that with greater *N*, more peaks in the ground truth are being fragmented but this benefit is rapidly cancelled out by a fast increase in the number of false positives. Similarly, recall increases with *N* initially but decreases (Figure 5b), suggesting that with greater *N*, more precursor ions from ground truth MS1 features are fragmented—up to the point when all possible precursor ions above the minimum intensity threshold of $1.75\times 10^{5}$ are selected. We can explain this trade-off between precision and recall due to the fact that as fragmentation coverage increases (with greater *N*), fewer ground truth peaks are detected from the fragmentation files (Figure 5c). The quality of MS1 chromatographic peak shapes in the fragmentation file becomes poorer since more duty cycle time is spent performing MS2 than MS1 scans, reducing the number of good-quality MS1 features that can be found by XCMS from the fragmentation files. Assessing the $F_{1}$ score (Figure 5d), which is the harmonic average of precision and recall and is representative of overall fragmentation performance, we see that the best $F_{1}$ score can be found at N=10. This is the same as the actual value of *N* used to generate the data (N=10) obtained by expert judgement. The results here demonstrate how a simulated environment such as ViMMS can be used to quantify the trade-off between fragmentation coverage and peak picking performance.

### 3.4. Varying Multiple Parameters in Top-N Simulations

To validate the use of ViMMS for Top-*N* method development, we now show how ViMMS compares to data generated at a wide range of *N* and DEW times. In the previous Section 3.3 DEW is fixed to 15 s for all values of *N*; however our hypothesis is that the best fragmentation performance can be obtained by optimizing both parameters simultaneously. Here we evaluate the ability of ViMMS to suggest the parameter combinations that provide the best fragmentation performance and compare the results to actual experimental data.

To validate simulated results, we generated a large real dataset in which the same sample, *BeerQCB* (introduced in Section 2.1) was fragmented using all combinations of N=(1,2,3,4,5,10,15,20,35,50) and DEW=(15,30,60,120). The minimum MS1 intensity threshold to fragment was completely disabled for this experiment to allow a consistent number of MS scans to be acquired under the different scenarios (see Section 2.1.3). To generate simulated data in ViMMS, we extracted ROIs from a full-scan MS1 analysis of the *BeerQCB* sample using the Existing Sample Workflow in Figure 1. These ROIs were used as input to the Top-*N* controller using the same ranges of parameters for *N* and DEW as the real data. For evaluation, peak picking using XCMS was performed on the full-scan and fragmentation mzML files, and fragmentation performance was computed on both real and simulated data following Section 3.3.

Inspecting parameter combinations in the heatmaps of Figure 6 we see a high level of agreement between the performance obtained from the simulated data, and that obtained from the real measurements. Optimal performance is observed in both cases for N=20 and DEW=30s although regions of high performance for both real and simulated results can be found at N=(10,15,20) and DEW=(15,30,60). Ranges of *N* that are either too large or too small demonstrate decreased performance in Figure 6a,b. Please note that the difference in optimum value in this experiment when compared with the previous one is explainable due to the use of a different MS platform (Q-Exactive orbitrap versus Fusion Tribid orbitrap).

Overall our findings demonstrate that ViMMS can be used to optimize Top-*N* acquisition methods in silico before actually running the experiment on a real MS instrument—something of great benefit to the community. Additional results are given in Appendix A.

### 3.5. DsDA Simulations

Finally, we show how ViMMS can be used to benchmark fragmentation strategies that work on multiple samples, such as DsDA [4] (Section 2.3.3). To benchmark DsDA using ViMMS, we generate synthetic data where samples are almost identical using the Synthetic Sample Workflow in Figure 1. To do this, 6500 chemical objects are generated by sampling formulae from HMDB (the *multi-beer* data is used to construct the spectral feature database). 20 samples are created from these chemical objects by adding independent Gaussian noise (with standard deviation set to 10,000) to the maximum intensities of the chemicals in the original sample. These 20 samples will have peaks in the same RT and *m*/*z* locations but with a slight variation in how intense they are. We compare the results from DsDA, DsDA MaxDepth and Top-4 DDA fragmentation strategies (N=4 was chosen as that is the default option for DsDA). Following the original DsDA study, performance is evaluated in terms of how many of the aligned peaks found by XCMS are successfully fragmented above a minimum intensity of $1.75\times 10^{5}$. Our experiment shows that DsDA and DsDA MaxDepth clearly outperform Top-4 DDA strategy in terms of how many chemicals they successfully fragment (Figure 7a). This is consistent with the results from the original DsDA study.

As a further investigation, we also compare the methods on the *multi-beer* and *multi-urine* data using the Existing Samples Workflow in Figure 1. ROIs are extracted from the full-scan mzML files of the two datasets and converted into chemical objects allowing us to virtually re-run the data under the DsDA fragmentation strategy using real chromatographic peaks. The result in Figure 7b,c shows that unlike previous results on synthetic data, here Top-4 DDA fragmentation strategy clearly gives the best performance in fragmenting the most peaks picked by XCMS, and no difference can be observed between DsDA and DsDA MaxDepth. Since DsDA prioritizes precursor peaks to fragment in a run based on previously seen runs, we explain the results here by the fact that the beer and urine samples are not similar enough for the DsDA strategy to be effective.

To confirm this, we return to our synthetic data and investigate the performance of the different methods as increasing numbers of chemicals are randomly removed from each sample. We consider scenarios where we randomly remove 0%, 5%, 10%, 15%, 20%, 25%, 30%, 35%, 40%, 45%, 50% of chemicals from each samples, meaning that on average samples will become less similar. In these samples, a given chemical object will appear in any two samples with a probability of 1, 0.90, 0.81, 0.72, 0.64, 0.56, 0.49, 0.42, 0.36, 0.30 and 0.25, respectively. In all cases, we generate 5 samples to run through the DsDA analysis. Figure 8 shows the number of chemicals fragmented above a minimum intensity of 1.75E5 after all five samples are processed by both the DsDA and Top-4 DDA fragmentation strategies in the different scenarios. The results show that DsDA performs well when the samples are similar, but as the samples becomes less similar the performance drops and DsDA is comfortably outperformed by the Top-4 DDA fragmentation strategy. Hence, as samples become more different, a Top-4 strategy should be preferred, but where samples are very similar (e.g., technical replicates), DsDA is likely to be more efficient.

Such experiments would be very challenging to do in reality. This example demonstrates how ViMMS can provide insight into the scenario in which a certain fragmentation strategy will be successful.

## 4. Discussion and Conclusions

In this paper, we introduce ViMMS, the first simulator specifically targeted at mass spectrometry fragmentation-based metabolomics that is modular, easily extensible, and can be used for the development, testing, and benchmarking of different fragmentation strategies. Processing MS2 data (particularly identifying spectra) is generally considered to be more challenging in metabolomics than in proteomics [28]. An in silico simulator such as ViMMS, which can be used to generate realistic-looking full-scan and fragmentation spectra based on either existing data or by sampling from a database of known metabolites, can be used to alleviate this problem. In this work, our experiments show how our proposed simulator can be used to help optimize acquisition methods in silico through two examples: Top-*N* DDA fragmentation, and DsDA. It is also important to note that our simulator could be used to create datasets on which novel data processing methods could be benchmarked.

The results from our experiments show that the spectral data generated from ViMMS have a strong resemblance to data produced from MS instruments. Our experiments with the Top-*N* and DsDA controllers in Section 3.2, Section 3.3, Section 3.4 and Section 3.5 demonstrate that despite some minor differences in output, the proposed simulator framework can be useful in investigating, understanding and comparing the characteristics of different fragmentation strategies. Furthermore, we provide insights in when best to use Top-*N* and DsDA fragmentation methods; something that is not that easily and cheaply done using experimental data.

When developing acquisition methods, selecting the *N* that provides the highest fragmentation performance and number of detected peaks can be challenging, particularly in the typical scenario where the full-scan data is assumed to be absent and peak picking quality from fragmentation files is therefore important for subsequent analysis. We demonstrated how ViMMS can be used to suggest *N* for use for similar future samples on the MS instrument. Our results show how ViMMS can be used to explore parameter combinations for a particular fragmentation strategies in silico for existing data, virtually re-run existing data under an alternative strategy and benchmark existing fragmentation strategies (like DsDA) with minimal modifications under the proposed framework. This is a capability not available from other simulators [5,6,7,8,9,10]. On top of fragmentation data, ViMMS can also be used to benchmark and perform comparative evaluation of different LC-MS data processing algorithm, such as peak picking and retention time alignment [29] in a more controlled manner. In each of these cases, ViMMS also has the potential to help develop new methods by allowing them to be evaluated in a scenario where the ground truth is known, and little machine time needed.

The modular nature of ViMMS means that as future work, we can extend it with different and improved noise models and test noise reduction approaches, additional improvement to MS1/MS2 spectral data generations through incorporating fragmentation spectra prediction methods such as CFM-ID [24,25] or NEIMS [26], as well as retention time predictions from chemical structures [15]. Expanding the capabilities in ViMMS by us or others (all code is open source) in the future will further enhance its utility. The target users of ViMMS are currently algorithmic and LC-MS/MS method developers. ViMMS is available as a Python package that can be accessed from Python scripts and interactive environments such as Jupyter Notebook from where users can point to their own spectral files or compound lists to start using ViMMS on their own data. However, for end-users who are not comfortable with scripting, we aim to build an easy-to-use graphical user interface on top of ViMMS. Finally, we plan to use the proposed framework to develop and evaluate novel model-based fragmentation strategies that produces the highest coverage of MS1 and MS2 fragmentation in real time.

## Figures and Tables

**Figure 1 metabolites-09-00219-f001:**
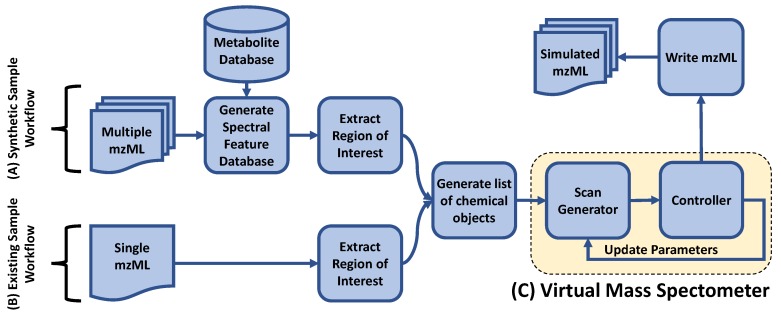
An overall schematic of ViMMS. (**A**) Synthetic Sample Workflow: Chemical objects in ViMMS can be created by sampling for compound formulae, mz, RT, and intensity values from the spectral feature database. (**B**) Existing Sample Workflow: alternatively, chemical objects can be created by extracting regions of interest from a single mzML file and converting them to chemical objects. (**C**) Yellow box: chemical objects are processed in the virtual mass spectrometer during in silico scan simulations. A controller performs parameter updates on the mass spectrometer depending on the fragmentation strategy implemented in the controller. Simulated results can be written as mzML files.

**Figure 2 metabolites-09-00219-f002:**
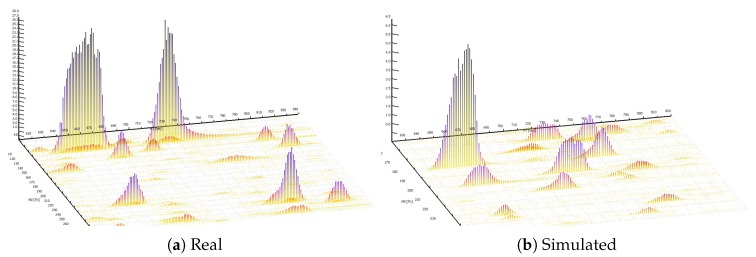
Real and simulated example outputs. (**a**) A region from the *beer-multibeers-1* LC/MS data. (**b**) A region from an LC/MS datafile generated by randomly generating peaks (mz, RT, intensity, chromatographic shape) from a database of peaks extracted from all *multi-beer* data.

**Figure 3 metabolites-09-00219-f003:**
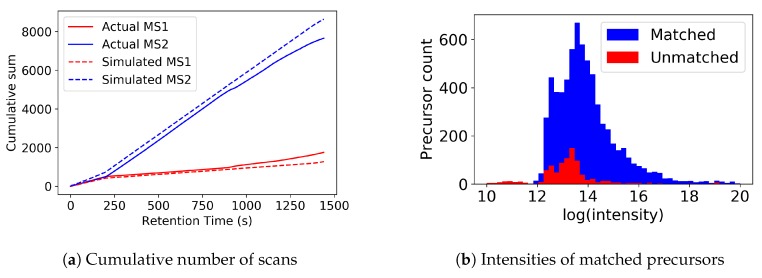
Figures showing (**a**) the cumulative number of MS1 and MS2 scans over time for real and simulated data, and (**b**) matched precursors from the actual *multi-beer-1* data to the simulated data. Most precursors that could be matched (blue) have higher intensities than those that cannot be matched (red).

**Figure 4 metabolites-09-00219-f004:**
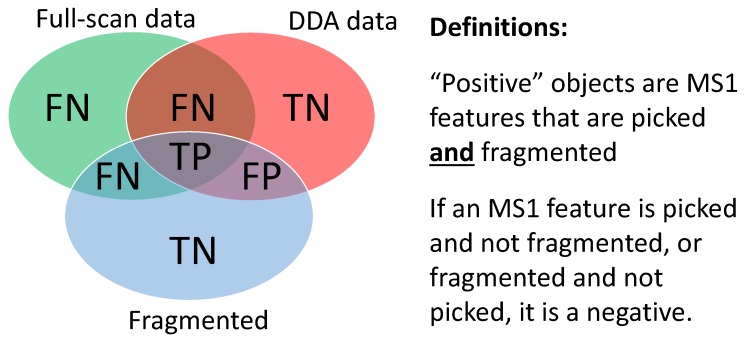
Definitions of True Positives (TP), False Positives (FP), True Negatives (TN) and False Negatives (FN) for performance evaluation of Top-*N* DDA fragmentation strategy. The blue circle in the Venn diagram refers to all MS1 features that are fragmented above the minimum MS1 intensity threshold, the green circle refers to all MS1 features found by XCMS’ CentWave from the full-scan file, while the red circle refers to all MS1 features found by CentWave from the fragmentation file.

**Figure 5 metabolites-09-00219-f005:**
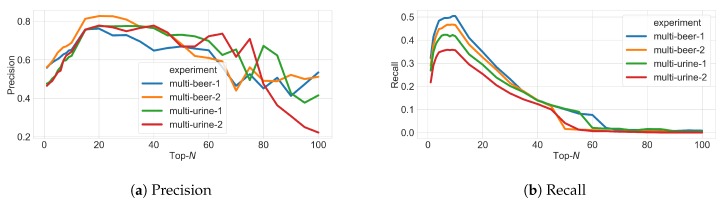
Figures showing (**a**) precision, (**b**) recall, (**c**) the number of peaks picked, and (**d**) F${}_{1}$ score for peak picking performance as *N* changes in Top-*N* DDA experiments in ViMMS based on the classification specifications given in Figure 4.

**Figure 6 metabolites-09-00219-f006:**
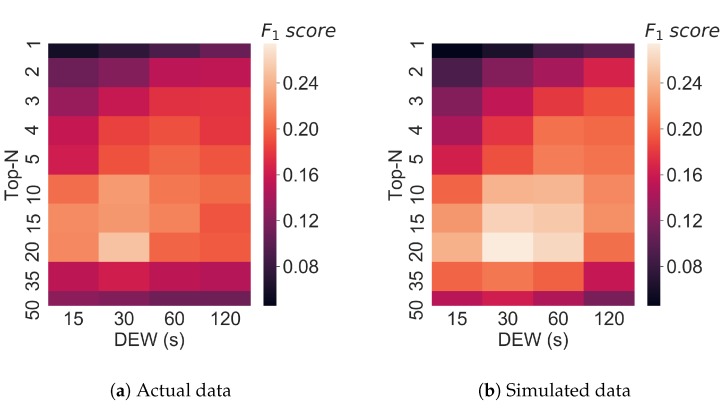
Fragmentation performance in terms of F${}_{1}$ score for (**a**) an actual *BeerQCB* sample, (**b**) simulated results from ViMMS.

**Figure 7 metabolites-09-00219-f007:**
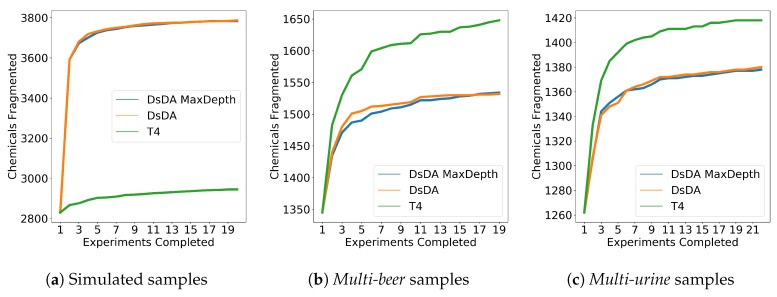
Top 4, DsDA and DsDA MaxDepth performance for in terms of the number of chemicals fragmented for (**a**) the simulated samples, (**b**) the *multi-beer* samples and (**c**) the *multi-urine* samples.

**Figure 8 metabolites-09-00219-f008:**
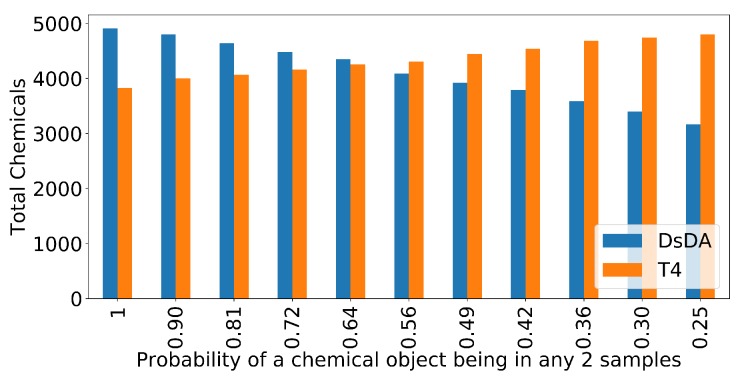
DsDA and Top 4 DDA performance in terms of the number of chemicals fragmented over multiple simulated datasets with varying dropout. In each scenario a percentage of chemicals are dropped from the sample (0%, 5%, 10%, 15%, 20%, 25%, 30%, 35%, 40%, 45%, 50%), meaning that on average samples will become less similar. In these samples, a given chemical object will appear in any two samples with a probability of 1, 0.90, 0.81, 0.72, 0.64, 0.56, 0.49, 0.42, 0.36, 0.30 and 0.25, respectively.

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
