# Peer review of "In Silico Optimization of Mass Spectrometry Fragmentation Strategies in Metabolomics"

_metabolites, 2019, doi:10.3390/metabo9100219_

Round 1

Reviewer 1 Report

In my opinion the presented manuscript is suitable for publication in Metabolites journal, however, I suggest to list more details concerning the software and scripts used i.e. “algorithm originally available in R” (line 173) could suggest that it means R Project software (GNU), but it is only my supposition.    

Author Response

We thank the reviewer for the comments. We have added a paragraph in the supplementary document (Section 5) describing the implementation in more detail and have properly referenced software packages where they are used in both the main text and the supplement. Changes to the manuscript have been highlighted in red.

Reviewer 2 Report

The authors present a framework to optimize MS/MS data acquisitions for LC-MS/MS based metabolic profiling. The authors developed the framework and validated it with experimental data.

Major issues:

1) P2L37: It would be important to add that metabolic profiling experiments are also harder to evaluate, because no sequence database exist and therefore FP/FN rates and decoys are hard to evaluate. I would write one or two additional sentences how it is different and why it is complex. Reason, proteomics approaches always are presented as easy, elegant and work with large number of spectra and IDs; that is unfortunately not the case in metabolomics.

2) I would love to see some tables in the main text with peak numbers or metabolite numbers. That could include known and unknown compounds. In blood plasma we can see around 1000 known metabolites (Metabolon) and maybe 2-3000 additional unknown compounds or more. So whatever the software simulates and how complicated it is, I want to know: Does it improve the coverage for the practitioner in the lab?

3) While the text and supplement and code is interesting, the current paper contains too many figures and discussions that are tailored towards bioinformatics users. However as I understand the target audience are from the field of mass spectrometry and metabolomics. So here the outcomes and improvements and maybe also some structural examples of additional compounds detected would be worthwhile.  I am recommending to think about figures that would create excitement for mass spectrometrists and metabolomics researchers.

 4) I am not a fan of peak picking or features, when we actually should talk about compounds or deconvolution. So peak picking was 10-15 years ago and we are now in year 2019. So a figure or discussion that shows 20,000 peaks (figure 5) is something I actually don’t like to see. A 15 minute HILIC run should have 1000-2000 compounds and adducts should be correctly assigned.

5) Is there a way for real-time adduct detection, real-time compound deconvolution (MS1 and MS2) based or is it just really an iterative or recursive process that puts ions into an exclusion list and lowers the threshold for data acquisition.

6) The abstract contains too much technicalities.  I would shorten it to 150 words and similar to a 20 second elevator talk/pitch describe what it does, what its good at and why everybody in the field should use it.

Minor issues:

1) P2L42: Please add the poster “Real-Time Instrument Control of the

Orbitrap Tribrid Mass Spectrometer” as a reference, poster or application note with full title and authors. Links are known to disappear quite quickly.

2) It would be worthwhile to add relevant references that use the Fusion API such as:

Full-featured, real-time database searching platform enables fast and accurate multiplexed quantitative proteomics ; Schweppe et al; BioArxiv https://www.biorxiv.org/content/10.1101/668533v1.full

3) ADAP is a recently developed peak picking and deconvolution algorithm for xcms and MZMIne. It would be interesting to know if ADAP improves the results or not.

4) Add known issues, weaknesses and potential improvements.

Author Response

We thank the reviewer for the comments. Our responses are provided below, and changes to the manuscript have been highlighted in red.

Comments and Suggestions for Authors

The authors present a framework to optimize MS/MS data acquisitions for LC-MS/MS based metabolic profiling. The authors developed the framework and validated it with experimental data.

We thank the reviewer for the comments. We’d like to make one general comment in response (as well as individual ones below). Our aim in this work has been the development of a simulator that makes it possible to optimise MS/MS acquisition in-silico, without having to use up machine time. As such, we are not claiming that we have produced a new acquisition strategy, but rather a tool that would make such an acquisition strategy easier to optimise. We have made this clearer in the introduction.

Major issues:

1) P2L37: It would be important to add that metabolic profiling experiments are also harder to evaluate, because no sequence database exist and therefore FP/FN rates and decoys are hard to evaluate. I would write one or two additional sentences how it is different and why it is complex. Reason, proteomics approaches always are presented as easy, elegant and work with large number of spectra and IDs; that is unfortunately not the case in metabolomics.

We have broadened the discussion to include comments as to why analysis of MS2 fragmentation data is more challenging in metabolomics than in proteomics.

2) I would love to see some tables in the main text with peak numbers or metabolite numbers. That could include known and unknown compounds. In blood plasma we can see around 1000 known metabolites (Metabolon) and maybe 2-3000 additional unknown compounds or more. So whatever the software simulates and how complicated it is, I want to know: Does it improve the coverage for the practitioner in the lab?

We agree that at the moment, the majority of metabolites measured in untargeted studies is a small proportion of the total. Our simulator can be populated either from a pre-existing MS1 file (in which case the complexity is identical to real data) or by drawing chemical formulas randomly from a database (e.g. HMDB). In the latter, users can sample as many formulas as they like.

3) While the text and supplement and code is interesting, the current paper contains too many figures and discussions that are tailored towards bioinformatics users. However as I understand the target audience are from the field of mass spectrometry and metabolomics. So here the outcomes and improvements and maybe also some structural examples of additional compounds detected would be worthwhile.  I am recommending to think about figures that would create excitement for mass spectrometrists and metabolomics researchers.

As described above, our aim here was the development of a simulator that makes optimisation of acquisition easier. In that light, we do not have examples of new structures.

4) I am not a fan of peak picking or features, when we actually should talk about compounds or deconvolution. So peak picking was 10-15 years ago and we are now in year 2019. So a figure or discussion that shows 20,000 peaks (figure 5) is something I actually don’t like to see. A 15 minute HILIC run should have 1000-2000 compounds and adducts should be correctly assigned.

We completely agree that adducts (and other products of the same molecular species, e.g. in-source fragments) should be correctly assigned. This is a challenging problem, and one on which we’ve worked in the past. In our experience, many users still pick chromatographic peaks with tools such as XCMS and MZmine. As such, we use picked peaks to enable us to evaluate the effect on peak picking performance of fragmentation (in particular, increasing N in a top-N strategy). Deconvolution in the context of MS2 data is relevant for DIA acquisition and our simulator would provide an excellent route for evaluating DIA deconvolution strategies. We have added an additional comment about this in Section 3.3.

5) Is there a way for real-time adduct detection, real-time compound deconvolution (MS1 and MS2) based or is it just really an iterative or recursive process that puts ions into an exclusion list and lowers the threshold for data acquisition.

Acquisition strategies such as these (that label adducts etc in real time) could be easily implemented within the modular simulator of ViMMS, and evaluated and compared against alternatives. This is something that has been lacking from the community, and it is an avenue that we have considered for future work.

6) The abstract contains too much technicalities.  I would shorten it to 150 words and similar to a 20 second elevator talk/pitch describe what it does, what its good at and why everybody in the field should use it.

We have modified the abstract to make the contribution clearer (P1 L9-11).

Minor issues:

1) P2L42: Please add the poster “Real-Time Instrument Control of the

Orbitrap Tribrid Mass Spectrometer” as a reference, poster or application note with full title and authors. Links are known to disappear quite quickly.

We have modified the reference to include this.

2) It would be worthwhile to add relevant references that use the Fusion API such as:

Full-featured, real-time database searching platform enables fast and accurate multiplexed quantitative proteomics ; Schweppe et al; BioArxiv https://www.biorxiv.org/content/10.1101/668533v1.full

We have added this to the initial discussion, describing where the fusion API has been used elsewhere.

3) ADAP is a recently developed peak picking and deconvolution algorithm for xcms and MZMIne. It would be interesting to know if ADAP improves the results or not.

As described above, peak picking in this case is used as part of the RoI extraction and evaluation processes. It would be straightforward to swap the XCMS alignment / peak picking that we’re currently using with MZmine. We have made this clear in Sections 2.2.4 and 3.1.

4) Add known issues, weaknesses and potential

We have broadened the discussion to better cover these areas.